# Research on the influence factors of sustainable development of plateau characteristic agriculture based on DEMATEL and AISM combined model

Wei Wang[◉], Hai Liu[iD]*[◉], Pengfei Zhao, Mo Han

National Engineering Research Center for Information Technology in Agriculture, Beijing Academy of Agriculture and Forestry Sciences, Beijing, China

◉ These authors contributed equally to this work.
* liuh@nercita.org.cn

**Data Availability Statement:** All relevant data are within the paper and its Supporting information files.

## Abstract

Under the background of the continuous progress of China's agricultural reform, the development of characteristic agriculture is an important field of agricultural development in the country and even the world. Yunnan has unique advantages in geography, location, climate, and human resources, and has unique conditions for the development of agriculture with plateau characteristic. However, the sustainable development of agriculture with plateau characteristic is affected and restricted by many factors. How to promote the sustainable development of agriculture with plateau characteristic is an important research topic, and it is also the main purpose of this study. Therefore, through literature analysis and investigation, this study studies the advantages and disadvantages, development status, main problems, countermeasures and suggestions, macro development direction, and theoretical research methods of characteristic agriculture in the Yunnan plateau. This paper analyzes the advantages and disadvantages of the Decision-making Trial and Evaluation Laboratory (DEMATEL) and the Adversarial Interpretation Structure Modeling Method (AISM) and proposes to combine the two models to make up for each other and improve the accuracy of model analysis. On this basis, a set of identification and evaluation systems of factors affecting the sustainable development of agriculture with plateau characteristic is established, which can comprehensively and accurately identify and evaluate various influencing factors, and provide a scientific basis for the sustainable development of agriculture with plateau characteristic. Finally, combined with the relevant statistical data of Yunnan Province from 2009 to 2020 and many results calculated by the model, the factors affecting the sustainable development of agriculture with plateau characteristic were comprehensively analyzed. The results show that transportation, environment, and insurance support factors are the root factors that affect the sustainable development of agriculture with plateau characteristic, while the regional economy, scientific and technological support, and the development of agricultural enterprises are of high importance. Financial support, the level of agricultural mechanization, the quality of labor, and other factors can not be ignored. Finally, according to the results of the analysis, the corresponding policy recommendations are put forward to

**Funding:** This work was supported in part by the Major Project of Science and Technology of Yunnan Province 202002AE090010. The funders had no role in study design, data collection and analysis, decision to publish, or preparation of the manuscript.

**Competing interests:** The authors have declared that no competing interests exist.

provide a reference for the sustainable development of plateau characteristic agriculture in Yunnan Province.

## 1. Introduction

With the continuous promotion of national agricultural reform, Chinese agriculture has entered a critical period of conceptual change, mechanism innovation, structural transformation and industrial reshaping [1, 2]. The development of special agriculture has always been an important area of agricultural development in the country and even in the world. Yunnan is endowed with unique advantages in geography, location, climate, and human resources. Agriculture has been in a dominant position in the socio-economic development of Yunnan, and its advantages and disadvantages are very obvious. On the one hand, it has been small, scattered, weak, and poor due to the mountainous geographical conditions that restrict the development of scale and mechanization; on the other hand, its resource endowment and natural climatic advantages make it very suitable for the development of specialty agriculture [3–5].

In various works of literature, domestic scholars have studied the current research on the development of plateau characteristic agriculture in Yunnan mainly from the perspectives of brand building, industrialization, agricultural ecology, development subjects, and government agricultural input policies, targeting its development advantages and disadvantages, development status, main problems, countermeasures and suggestions, macro development directions, and theoretical elaboration. Kong et al. [6], analyzed the advantages and conditions for the development of plateau characteristic agriculture in Yunnan, summarized the effectiveness and experience of development, and proposed the overall ideas and measures for development. Wu et al. [7] studied the development strategy of Yunnan plateau characteristic agriculture and proposed to give full play to the synergistic function of holistic and regional strategies to promote agricultural production, farmers' income, and rural prosperity, and contribute to the promotion of the province's economic and social leapfrog development. Dong et al. [8] used a combination of qualitative and quantitative methods to select four categories of qualitative and six categories of quantitative indicators with 32 Chinese provinces (municipalities and autonomous regions) as quantitative reference objects and concluded that diversity, ecological security, and time difference complementarity are the comparative advantages of Yunnan's plateau characteristic agricultural development. Long et al. [9] concluded that Yunnan plateau characteristic agriculture should adopt a target market strategy to enhance the value of agricultural products and promote a virtuous cycle for the development of Yunnan plateau characteristic agriculture. Xiang et al. [10, 11] studied the branding and positioning of Yunnan plateau characteristic agricultural products and pointed out that the core of the brand should be "health", and based on the analysis of the advantages and disadvantages of brand development, they pointed out that the branding of Yunnan plateau characteristic agricultural products should be developed into functional models such as culture, tourism, and technology under the premise of industrial cluster model. Deng et al. [12] reviewed the current situation, main problems, and countermeasure suggestions for the development of electronic trading of bulk agricultural products with plateau characteristic in Yunnan. At present, domestic research on Yunnan plateau characteristic agriculture mainly focuses on the development strategy, problems and countermeasures, marketing strategy, brand building, and comparative advantages of Yunnan plateau characteristic agriculture, and lacks research on the plateau characteristic agricultural resources and their distribution in Yunnan province. The above-mentioned research areas cover government public functions, business management, financial services, science and

technology research and development, ecological protection, and sustainable development. The research covers a relatively scattered scope, and much of the research literature is based on relatively more qualitative discussions and less quantitative research [13–20]. It is an important work to promote the sustainable development process of plateau characteristic agriculture by analyzing the relationship and intensity of influence of each factor on the sustainable development of plateau characteristic agriculture and constructing an evaluation system for the sustainable development of plateau characteristic agriculture after considering all factors, so as to clarify the problems in the sustainable development of plateau characteristic agriculture in each region.

In order to help decision-makers analyze the problem more comprehensively, some scholars at home and abroad try to use quantitative analysis and mixed analysis methods to make more scientific and effective decisions. Zhu et al. [21]. Based on the survey of 445 cotton growers in Xinjiang, and users DEMATEL (Decision-making Trial and Evaluation Laboratory, DEMATEL) and ISM (Interpretative Structural Modeling Method, ISM) integrated analysis method to identify the key factors that affect the deviation of green ecological agricultural technology application intention and behavior. Xia et al. [22] used the Grey-DEMATEL method to identify causal factors, influencing factors, and central factors of the key obstacles to the development of circular agriculture by the government, farmers, and enterprises, and described the correlation among the obstacles. Barbosa et al. [23] used the fuzzy DEMATEL method to identify the obstacles affecting the development of sustainable agriculture and analyzed which obstacles were more influential and vulnerable. Khan et al. [24] aimed to use the improved Delphi method, the Best-Worst Method (BWM), and interpretive structural modeling (ISM) to analyze supply chain factors affecting public sector agricultural development projects in Bangladesh and to determine the relationships among all factors.

From the perspective of research methods, both DEMATEL and ISM models are decision-support tools for analyzing complex problems. DEMATEL determines the core factors by analyzing the relationship between the problem factors, while ISM determines the core factors by analyzing the hierarchical structure and causal relationship between the problem factors. The improved AISM (Adversarial Interpretive Structure Modeling Method, AISM) model based on the ISM model is a decision support model based on adaptive network analysis, which has higher accuracy, stronger adaptability, and a wider application range than the ISM model. It can adaptively adjust the network structure, use a new weight calculation method to evaluate the importance of factors, and support a variety of decision analysis methods. The combination of the DEMATEL model and AIMS model can understand the problem more comprehensively, help decision makers to make more scientific and effective decisions, and provide a new idea for the study of the factors affecting the sustainable development of agriculture with plateau characteristic.

Based on clarifying the basic connotation of sustainable development of agriculture with plateau characteristic, this paper comprehensively considers the current situation and evaluation objectives of agriculture with plateau characteristic and selects relevant indicators from many levels to carry out research. This paper determines the index system that affects the sustainable development of agriculture with plateau characteristic, quantitatively identifies the evaluation of the sustainable development of agriculture with plateau characteristic by using the DEMATEL-AISM method, and makes clear the mutual influence and hierarchical relationship among the factors affecting the sustainable development of agriculture with plateau characteristic. Finally, the paper analyzes the relevant development of Yunnan Province in vigorously developing modern agriculture with plateau characteristic since 2012, to provide a theoretical basis and reference for the sustainable development policy formulation and development planning of plateau characteristic agriculture in Yunnan Province.

## 2. Overview of DEMATEL-AISM methods

DEMATEL (Decision-making Trial and Evaluation Laboratory, DEMATEL) is a commonly used factor analysis method to study the influence degree of complex systems, especially for systems with uncertain factor relationships [25, 26], but it lacks a quantitative basis for the determination of relationship strength values, and sometimes it is difficult to effectively integrate subjective Opinions vary widely. AISM (Adversarial Interpretive Structure Modeling Method, AISM) is also one of the effective techniques for complex system structure modeling. It is based on the classic Interpretive Structure Model (ISM) and integrated into the Adversarial idea in the Generative Adversarial Network (GAN). It is a newly proposed modeling method [27, 28]. AISM is good at expressing the structural relationship between elements, but it is inconvenient to deal with non-Boolean matrices, and it is also insufficient for the degree of influence relationship between elements.

In general, the DEMATEL model and the AISM model are two commonly used models for analyzing influencing factors. Both models have their advantages and disadvantages. The DEMATEL model is not accurate enough to evaluate the importance of influencing factors, while the AISM model is not clear enough about the interaction relationship between influencing factors. Therefore, combining the two models can reduce the calculation process, improve efficiency, make up for each other's shortcomings, and verify the accuracy of the model analysis [29–32]. Based on this, we can establish an identification and evaluation system for the factors affecting the sustainable development of plateau characteristic agriculture, identify and evaluate various influencing factors comprehensively and accurately, and provide a scientific basis for the sustainable development of plateau characteristic agriculture. The flow chart of the model is shown in Fig 1.

where O is the direct influence matrix, N is the normalized influence matrix, T is the combined influence matrix, D is the decision matrix composed of the absolute values of centrality and causality, A is the adjacency matrix calculated in the partial order, B is the phase multiplication matrix, R is the reachable matrix, R' is the reduced point reachable matrix, S' is the skeleton matrix, and S is the general skeleton matrix.

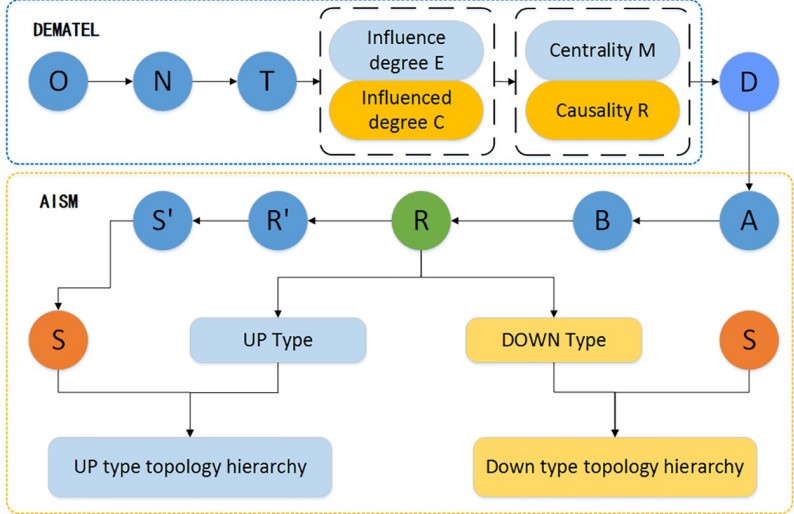

**Fig 1. The DEMATEL-ASIM model flow chart.**

# 3. Identification of influencing factors and construction of evaluation system

## 3.1. Influence factor identification and data acquisition

To select the impact factors of sustainable development of agriculture with plateau characteristic, the author read a large amount of literature, and after analyzing the existing literature and identifying word frequency clouds of 395 collated literature to screen out the word frequency clouds of impact factors, a total of 65 impact factors of sustainable development of agriculture with plateau characteristic were collated, and through a combination of field interviews and questionnaires from relevant research scholars and practitioners related to agriculture, forestry, animal husbandry, and fishery. Through merging and sorting, 30 impact indicators of sustainable development of plateau characteristic agriculture were finally selected, and the impact factor evaluation index system is shown in Table 1.

The above 30 influencing factors were refined and discussed by experts, deleted, combined and summarized into 18 factors; agricultural capital investment (A1), agricultural price index (A2, previous year = 100), labor force quality (A3, proportion of illiterate people over 15 years old), total agricultural machinery power (A4), transportation cargo turnover (A5), regional economic development level (A6), qualification rate of agricultural products (A7), number of agricultural enterprises and organizations (A8), level of mechanization of agricultural machinery (A9), agricultural labor productivity (A10), per capita net income of farmers (A11), total agricultural output value per hectare (A12), agricultural fertilizer application (A13), pesticide use per hectare (A14), financial support (A15), insurance support (A16, insured amount), brand building (A17) and technology support (A18), brand The statistical data of the relevant elements from 2009 to 2020 were obtained from the Yunnan Provincial People's Government, the National Bureau of Statistics of China and the China Green Food Development Center, and the results are shown in the following Table 2.

## 3.2. Influence factor modeling

**3.2.1. Determine the direct impact matrix O.** According to the elements in Table 2, combined with the results of the preliminary follow-up and questionnaire survey, the mutual

**Table 1. Statistics on factors affecting the sustainable development of plateau characteristic agriculture.**

| NO. | Factor name | NO. | Factor name |
|---|---|---|---|
| 1 | agricultural capital investment | 16 | agricultural production |
| 2 | regional economic development level | 17 | arable land area |
| 3 | rural informatization level | 18 | agricultural output |
| 4 | number of laborers | 19 | qualification rate of agricultural products |
| 5 | labor force quality | 20 | number of agricultural cooperatives |
| 6 | per capita net income of farmers | 21 | pesticide use per hectare |
| 7 | agricultural price index | 22 | technology support |
| 8 | agricultural labor productivity | 23 | agricultural fertilizer application |
| 9 | total agricultural output value per hectare | 24 | environmental subsidy policy |
| 10 | transportation cargo turnover | 25 | number of effective standard green food units |
| 11 | number of effective green food products | 26 | agriculture, forestry and water affairs inputs |
| 12 | agricultural production materials price index | 27 | agricultural insurance support |
| 13 | agricultural production price index | 28 | number of trademark applications |
| 14 | total agricultural machinery power | 29 | percentage of people with science and technology activities in independent research institutions |
| 15 | level of mechanization of agricultural machinery | 30 | number of agricultural enterprises and organizations |

**Table 2. Statistics on factors influencing the sustainable development of plateau characteristic agriculture from 2009 to 2020.**

| Item/Year | 2009 | 2010 | 2011 | 2012 | 2013 | 2014 | 2015 | 2016 | 2017 | 2018 | 2019 | 2020 |
|---|---|---|---|---|---|---|---|---|---|---|---|---|
| A1 (100 million yuan) | 267.28 | 327.21 | 409.8 | 518.6 | 538.97 | 594.45 | 641.52 | 712.92 | 674.82 | 842.2 | 1117.2 | 1100.13 |
| A2 (previous year = 100) | 98.8 | 103.1 | 104.9 | 102.0 | 101.4 | 101.0 | 100.1 | 100.7 | 101.1 | 101.9 | 102.0 | 101.4 |
| A3 (rate) | 13.74 | 11.59 | 8.71 | 8.34 | 8.45 | 8.23 | 9.53 | 8.83 | 8.39 | 8.14 | 7.31 | 5.78 |
| A4 (10000 kilowatts) | 2159 | 2411 | 2628.4 | 2874.5 | 3070.3 | 3215 | 3333 | 3440 | 3534.5 | 2694 | 2714 | 2787 |
| A5 (10000 tons) | 843.67 | 915.04 | 996.2 | 1092.09 | 1202.23 | 1407.63 | 1465.3 | 1569.2 | 1798.67 | 1943.85 | 1524.03 | 1551.07 |
| A6 (100 million yuan) | 6169.75 | 7224.18 | 8893.12 | 10309.47 | 12825.46 | 14041.65 | 14960 | 16369 | 18486 | 20880.63 | 23223.75 | 24521.9 |
| A7 (rate) | 78.9 | 89.3 | 81.3 | 87.78 | 89.97 | 93.86 | 95.86 | 95.5 | 93.89 | 93.95 | 92.05 | 92.5 |
| A8 (unit) | 7026 | 10100 | 14416 | 19175 | 23563 | 31186 | 54454 | 72197 | 103864 | 103864 | 85592 | 97737 |
| A9 (10000 kilowatts / 10000 persons) | 1.29 | 1.43 | 1.56 | 1.79 | 1.97 | 2.12 | 2.25 | 2.33 | 2.56 | 2.08 | 2.15 | 2.27 |
| A10 (100 million yuan / 10000 persons) | 1.02 | 1.07 | 1.37 | 1.66 | 1.96 | 2.15 | 2.29 | 2.51 | 2.81 | 3.17 | 3.92 | 4.83 |
| A11 (yuan) | 3369 | 3952 | 4722 | 5417 | 6724 | 7456 | 8242 | 9020 | 9862 | 10768 | 11902 | 12842 |
| A12 (100 million yuan) | 2.81 | 2.98 | 3.79 | 4.31 | 4.91 | 5.25 | 5.45 | 5.97 | 6.23 | 6.62 | 7.95 | 9.54 |
| A13 (10000 tons) | 171.39 | 184.58 | 200.47 | 210.21 | 219.02 | 227.01 | 231.33 | 235.58 | 231.94 | 217.37 | 204.03 | 196.65 |
| A14 (10000 tons) | 4.26 | 4.62 | 4.82 | 5.53 | 5.48 | 5.72 | 5.86 | 5.86 | 5.77 | 5.26 | 4.74 | 4.48 |
| A15 (100 million yuan) | 37.98 | 42.86 | 56.6 | 65.34 | 85.18 | 86.3 | 97.12 | 93.72 | 106.84 | 109.88 | 264.16 | 228.91 |
| A16 (100 million yuan) | 57.01 | 265.55 | 1174.31 | 1303.17 | 869 | 1217.77 | 1772.59 | 1085.18 | 1280.35 | 1382.14 | 1489.59 | 1533.41 |
| A17 (unit) | 209 | 221 | 181 | 195 | 213 | 235 | 261 | 288 | 301 | 384 | 501 | 567 |
| A18 (rate) | 0.83 | 0.82 | 0.84 | 0.82 | 0.83 | 0.82 | 0.85 | 0.84 | 0.83 | 0.85 | 0.86 | 0.86 |

influence relationship between the elements was assigned, and the degree of influence between the elements was divided into 5 levels and assigned 0 to 4 respectively, with increasing degrees of influence, to obtain the direct influence matrix, and the results are shown in the following Table 3.

**3.2.2. The calculation specification directly affects the matrix N.** The data were processed using the averaging method to obtain the normative direct impact matrix N. The formula is shown in Eq (1).

$$N = O / \max \sum\nolimits_{j=1}^{n} o_{ij} \tag{1}$$

Where O is the direct influence matrix.

**Table 3. Direct influence matrix.**

|      | A1 | A2 | A3 | A4 | A5 | A6 | A7 | A8 | A9 | A10 | A11 | A12 | A13 | A14 | A15 | A16 | A17 | A18 |
|------|----|----|----|----|----|----|----|----|----|-----|-----|-----|-----|-----|-----|-----|-----|-----|
| A1   | 0  | 0  | 0  | 3  | 0  | 3  | 0  | 3  | 3  | 0   | 0   | 0   | 0   | 0   | 3   | 3   | 3   | 3   |
| A2   | 0  | 0  | 0  | 0  | 2  | 0  | 0  | 0  | 0  | 0   | 3   | 0   | 0   | 0   | 0   | 0   | 0   | 0   |
| A3   | 0  | 0  | 0  | 0  | 0  | 2  | 3  | 0  | 2  | 2   | 3   | 2   | 2   | 2   | 0   | 0   | 2   | 2   |
| A4   | 0  | 1  | 0  | 0  | 1  | 2  | 0  | 0  | 4  | 4   | 1   | 4   | 0   | 0   | 0   | 0   | 0   | 0   |
| A5   | 0  | 3  | 0  | 0  | 0  | 3  | 0  | 1  | 0  | 0   | 2   | 0   | 0   | 0   | 0   | 0   | 3   | 0   |
| A6   | 3  | 1  | 3  | 3  | 3  | 0  | 0  | 3  | 0  | 0   | 3   | 0   | 0   | 0   | 3   | 1   | 3   | 3   |
| A7   | 0  | 1  | 0  | 0  | 0  | 0  | 0  | 0  | 0  | 0   | 2   | 0   | 0   | 0   | 0   | 2   | 4   | 2   |
| A8   | 0  | 0  | 3  | 1  | 2  | 3  | 0  | 0  | 1  | 0   | 3   | 0   | 0   | 0   | 1   | 1   | 3   | 4   |
| A9   | 0  | 2  | 0  | 4  | 0  | 2  | 2  | 0  | 0  | 4   | 3   | 4   | 0   | 0   | 0   | 2   | 0   | 2   |
| A10  | 0  | 2  | 0  | 0  | 1  | 2  | 0  | 0  | 0  | 0   | 3   | 4   | 1   | 1   | 0   | 0   | 2   | 0   |
| A11  | 0  | 0  | 3  | 2  | 0  | 3  | 0  | 0  | 1  | 1   | 0   | 0   | 1   | 1   | 0   | 0   | 2   | 0   |
| A12  | 0  | 4  | 0  | 0  | 2  | 3  | 0  | 2  | 1  | 0   | 3   | 0   | 0   | 0   | 0   | 3   | 0   | 2   |
| A13  | 0  | 1  | 0  | 0  | 0  | 0  | 1  | 0  | 0  | 3   | 2   | 2   | 0   | 0   | 0   | 0   | 2   | 2   |
| A14  | 0  | 1  | 0  | 0  | 0  | 0  | 4  | 0  | 0  | 3   | 2   | 3   | 0   | 0   | 0   | 1   | 3   | 2   |
| A15  | 4  | 0  | 0  | 4  | 2  | 0  | 2  | 3  | 3  | 0   | 1   | 1   | 0   | 0   | 0   | 0   | 1   | 3   |
| A16  | 0  | 0  | 0  | 0  | 0  | 0  | 0  | 0  | 0  | 0   | 3   | 0   | 0   | 0   | 0   | 0   | 0   | 0   |
| A17  | 0  | 3  | 0  | 0  | 0  | 3  | 3  | 3  | 0  | 0   | 3   | 1   | 3   | 3   | 0   | 0   | 0   | 0   |
| A18  | 0  | 0  | 2  | 3  | 0  | 0  | 3  | 3  | 1  | 3   | 2   | 2   | 2   | 2   | 0   | 0   | 3   | 0   |

**3.2.3. Calculate the integrated impact matrix T.** The integrated impact matrix T is obtained using the canonical direct impact matrix calculated as follows.

$$T = \left(N + N^2 + N^3 + \ldots + N^k\right) = \sum_{k=1}^{\infty} N^k = N(1 - N)^{-1} \tag{2}$$

Where N is the normalized influence matrix.

**3.2.4. Obtain the calculated metric values for the DEMATEL algorithm.** Using the integrated influence matrix to calculate the value of each indicator, the influence degree E value is calculated by Eq (3), the influenced degree C value is calculated by Eq (4), and the central degree M value is calculated by E + C, and the cause degree R value is calculated by E—C. Using the meaning of centrality and normalizing it to obtain the weight value of each element W. DEMATEL calculates the index values as follows, and the results are shown in the following Table 4.

$$E_i = \sum_{j=1}^{n} t_{ij}, (i = 1, 2, 3, \ldots, n) \tag{3}$$

$$C_i = \sum_{j=1}^{n} t_{ji}, (i = 1, 2, 3, \ldots, n) \tag{4}$$

$$M_i = E_i + C_i \tag{5}$$

$$R_i = E_i - C_i \tag{6}$$

Where T is the comprehensive influence matrix and t is the determinant value.

The results of the weight ranking (Fig 2) are as follows.

The weight value of each element is obtained by normalizing the 'center degree E+C value'. Fig 2 displays the weight value for each element, which represents the element's importance in the system. The higher the value, the more important the element.

**Table 4. Calculated index values for the DEMATEL algorithm.**

|  | E | C | E+C(M) | E-C(R) | W |
|---|---|---|---|---|---|
| **A1** | 2.455 | 0.593 | 3.048 | 1.862 | 0.051 |
| **A2** | 0.400 | 1.839 | 2.239 | -1.440 | 0.037 |
| **A3** | 2.046 | 1.335 | 3.380 | 0.711 | 0.056 |
| **A4** | 1.607 | 1.808 | 3.414 | -0.201 | 0.057 |
| **A5** | 1.124 | 1.261 | 2.385 | -0.137 | 0.040 |
| **A6** | 2.841 | 2.563 | 5.404 | 0.278 | 0.090 |
| **A7** | 0.937 | 1.572 | 2.508 | -0.635 | 0.042 |
| **A8** | 2.235 | 1.695 | 3.929 | 0.540 | 0.066 |
| **A9** | 2.130 | 1.342 | 3.472 | 0.788 | 0.058 |
| **A10** | 1.422 | 1.764 | 3.186 | -0.342 | 0.053 |
| **A11** | 1.448 | 3.606 | 5.054 | -2.158 | 0.084 |
| **A12** | 1.682 | 1.998 | 3.680 | -0.316 | 0.061 |
| **A13** | 1.152 | 1.018 | 2.170 | 0.134 | 0.036 |
| **A14** | 1.588 | 1.018 | 2.606 | 0.570 | 0.043 |
| **A15** | 2.399 | 0.626 | 3.025 | 1.773 | 0.050 |
| **A16** | 0.253 | 1.099 | 1.352 | -0.846 | 0.023 |
| **A17** | 1.913 | 2.790 | 4.704 | -0.877 | 0.078 |
| **A18** | 2.355 | 2.058 | 4.414 | 0.297 | 0.074 |

**3.2.5. Plotting the degree of influence—degree of being influenced.** Using the degree of influence as the horizontal coordinate and the degree of being influenced as the vertical coordinate, draw the influence-degree of being influenced diagram as shown in Fig 3.

The influence degree-affected degree diagram can be used to analyze the influence and affected relationship of the elements, and the significance of its four quadrants is described as follows:

The first quadrant represents a high influence degree E and a high influence degree C value, indicating a high factor influence degree and high influence degree.

The second quadrant represents a low influence degree E value and a high influence degree C value, indicating a low factor influence degree and high influence degree.

The third quadrant represents a low influence degree E and a low influence degree C value, indicating a low factor influence degree and low influence degree.

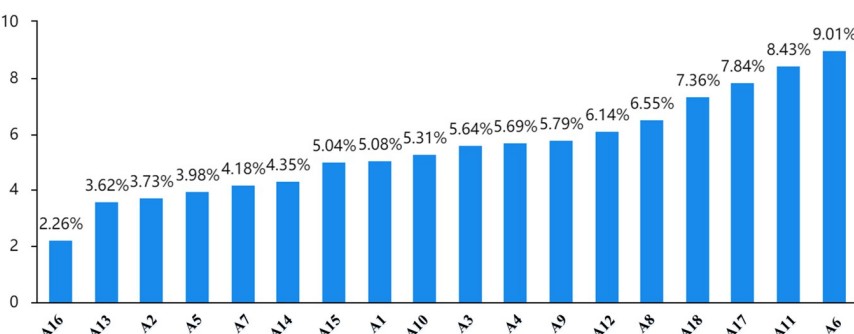

**Fig 2. The results of the weight ranking.**

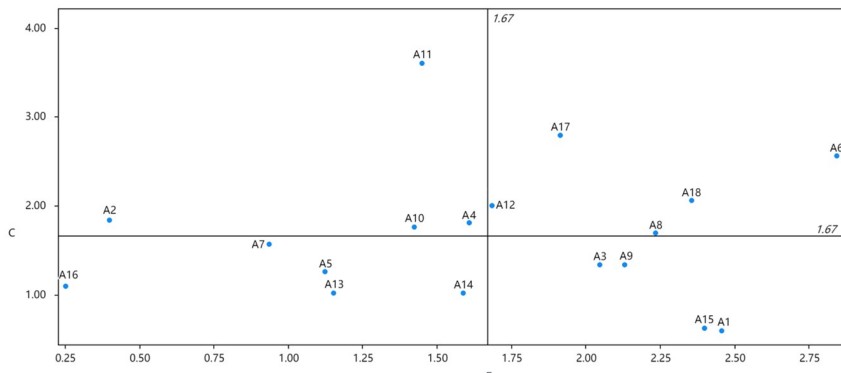

**Fig 3. The Influence degree—Influenced degree mapping.**

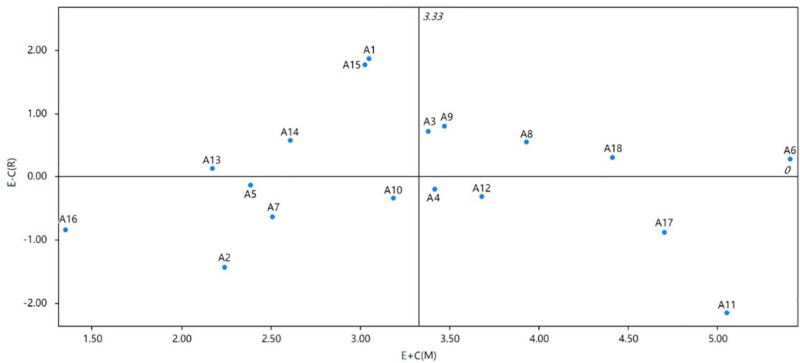

**Fig 4. The centrality-causality mapping.**

The fourth quadrant represents a high influence degree E value and a low influence degree C value, indicating a high factor influence degree and low influence degree.

**3.2.6. Plotting centrality-causality diagrams.** Using the centrality as the horizontal coordinate and the cause degree as the vertical coordinate, the cause result graph is drawn as shown in Fig 4.

The centrality-causality diagram analyzes the relationship between centrality and causality. The significance of the total four quadrants of the centrality-cause graph is as follows:

The first quadrant: both the center degree and the cause degree are high, that is, the elements are of high importance and are the cause factors.

The second quadrant: low center and high cause, that is, the importance of elements is low and is the cause factor.

The third quadrant: both the center degree and the cause degree are low, that is, the importance of the elements is low and is the resulting factor.

The fourth quadrant: a high degree of center and low degree of cause, that is, the importance of elements is high and is the result factor.

**3.2.7. Calculate the decision matrix D.** Based on the degree centrality of DEMATEL and the transitivity principle of influence, usually the absolute values of the center degree M and the cause degree |R| are first taken from the decision matrix D, and then the adjacency

matrix A is calculated by a partial order. The calculation method of decision matrix D is as follows:

$$D = [d_{ij}]_{n \times 2} \tag{7}$$

**3.2.8. Calculate the adjacency matrix A.**   Among them, the calculation rule for partial order is as follows: for any two rows x and y in the decision matrix D, the partial order relationship between elements x and y is recorded as x precedes y. The meaning of x elements is that y elements are superior to x elements. For the decision matrix D, the relation matrix A can be obtained using partial order rules. The partial order calculation can be done using formula (8), which is as follows:

$$D \xrightarrow{\preceq} A = [a_{ij}]_{n \times n} \tag{8}$$

$$a_{xy} = \begin{cases} 1, x < y \\ 0, x \geq y \end{cases} \tag{9}$$

**3.2.9. Calculate the reachable matrix R.**   The overall adjacency matrix B is obtained by calculating the adjacency matrix A with the following formula.

$$A \xrightarrow{A+I} B \tag{10}$$

After obtaining the overall adjacency matrix B, the matrix B is concatenated and multiplied to obtain the reachable matrix R, which is calculated as follows.

$$B = R^{k+1} = R^k \neq R^{k+1} \tag{11}$$

**3.2.10. AISM cascade extraction results.**   First, according to Eq (10) and the reachable matrix K, the reachable set R(ei), the prior set Q(ei) and the common set L(ei) of each factor Fi are established, and the adversarial hierarchy is divided by the result-first (division rule: L(ei) = R(ei)) and the cause first (division rule: L(ei) = Q(ei)) rules, respectively. Where the number represents an element, for example, 2 represents the 2nd element. The extraction result of Reachable sets and prior sets and their intersections is shown in Table 5.

A hierarchical adversarial division was performed to obtain the result-first and cause-first extraction results, as shown in the following Table 6.

Continue the point reduction of the reachable matrix R to obtain the reachable matrix R', and further perform the edge reduction operation on the matrix R' and bring in the loop factors to obtain the general skeleton matrix S. The calculation formula is as follows (I is the unit matrix).

$$S = R' - (R' - I')^2 - I \tag{12}$$

Finally, based on the association and extraction results among various factors, a schematic diagram of the directed topological hierarchy is drawn. The UP type and DOWN type directional topological hierarchy diagram is shown in Fig 5.

## 4. Results and analysis

### 4.1. Analysis of the root cause element set

As shown in Fig 5, the set of factors that are at the bottom of the system and are not influenced by other factors are the root cause factors, {A5,A13} ∪ {A5,A13,A16} = {A5,A13,A16},

**Table 5. Reachable sets and prior sets and their intersections.**

| Up type topology hierarchy extraction | | Down type topology hierarchy extraction | |
|---|---|---|---|
| reachable sets(R) | intersection set(L = R∩Q) | prior set(Q) | intersection set(L = R∩Q) |
| 1,11 | 1 | 1,2,5,7,13,14,15,16 | 1 |
| 1,2,11,15 | 2 | 2,5,13,16 | 2 |
| 3,11,17 | 3 | 3,5,7,10,13,14 | 3 |
| 4,6,8,9,11,12,17,18 | 4 | 4,5,13 | 4 |
| 1,2,3,4,5,6,7,8,9,10,11,12,13,14,15,17,18 | 5,13 | 5,13 | 5,13 |
| 6,11 | 6 | 4,5,6,13 | 6 |
| 1,3,7,9,11,15,17 | 7 | 5,7,13 | 7 |
| 8,11,17 | 8 | 4,5,8,10,12,13 | 8 |
| 9,11,17 | 9 | 4,5,7,9,10,13,14 | 9 |
| 3,8,9,10,11,17 | 10 | 5,10,13 | 10 |
| 11 | 11 | 1,2,3,4,5,6,7,8,9,10,11,12,13,14,15,16,17,18 | 11 |
| 8,11,12,17 | 12 | 4,5,12,13 | 12 |
| 1,2,3,4,5,6,7,8,9,10,11,12,13,14,15,17,18 | 5,13 | 5,13 | 5,13 |
| 1,3,9,11,14,15,17 | 14 | 5,13,14 | 14 |
| 1,11,15 | 15 | 2,5,7,13,14,15,16 | 15 |
| 1,2,11,15,16,17 | 16 | 16 | 16 |
| 11,17 | 17 | 3,4,5,7,8,9,10,12,13,14,16,17,18 | 17 |
| 11,17,18 | 18 | 4,5,13,18 | 18 |

transportation cargo turnover (A5), agricultural fertilizer application (A13) and insurance support (A16) are the set of root cause factors that can directly or indirectly influence the system within other factors. Root cause factors are dominant in the system, and their influence on the sustainable development of plateau characteristic agriculture is the most important.

At the same time, a loop exists for the set of resultant factors located at the bottom level. The loop is an important indicator of the rationality of the AISM modeling, and the fact that these two factors form a loop in the directed topological hierarchy diagram indicates that the model is mutually reachable and reasonable. The mutual accessibility relationship also indicates that transportation cargo turnover and agricultural fertilizer application are mutually influential relationships.

## 4.2. Analysis of the middle element set

The intermediate factor set, i.e., the set of factors that receive the underlying causes and send upward arrows to influence the upper factors, can be used as an expression of the intermediate

**Table 6. Result-first and reason-first extraction results.**

| Levels | Results first–type up | Reason first–type down |
|---|---|---|
| 0 | 11 | 11 |
| 1 | 1,6,17 | 17 |
| 2 | 3,8,9,15,18 | 1,8 |
| 3 | 2,7,10,12,14 | 3,6,9,12,15,18 |
| 4 | 4,16 | 2,4,7,10,14 |
| 5 | 5,13 | 5,13,16 |

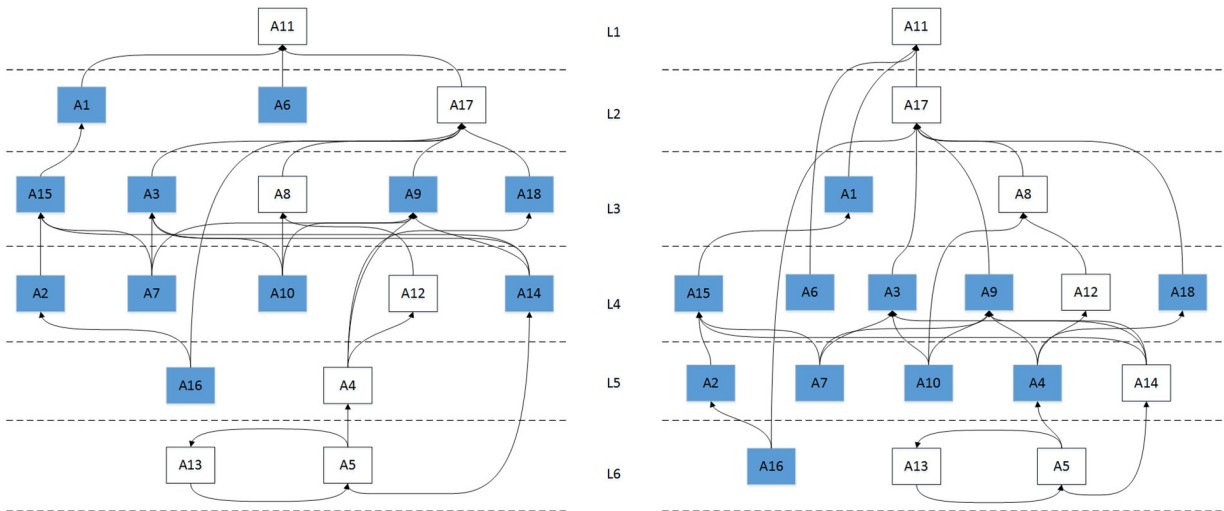

**Fig 5. Up type and down type directional topology hierarchy.**

state of system development, reflecting the state of system development. As shown in Fig 4, this factor set spans four levels. Two points deserve special attention.

First, five factors belong to the same L6 level: communication domain (S16), communication competence (S7), social status (S8), cultural adaptation (S9), and culture shock (S15). This suggests that there is an interrelated relationship between these five elements.

Second, the evaluation of the hierarchical abstraction process revealed that 11 activity elements experienced hierarchical jumps, 10 of which jumped between the L2->L5 layers, including agricultural capital input (A1), agricultural price index (A2), labor quality (A3), regional economic development level (A6), agricultural product qualification rate (A7), agricultural machinery mechanization level (A9), agricultural labor productivity (A10), pesticide use per hectare (A14), financial support (A15), and scientific and technological support (A18). In addition, there is 1 intermediate factor insurance support (A16) that jumps directly from the L5 level to the bottom level of the system, L6, as the root cause element. These 11 elements are active elements, thus indicating that the system is an active system, i.e. a topologically variable system. In addition, it can be seen from the second table that the system has an unstable and variable character, influenced by the 11 elements mentioned above.

## 4.3. Impact degree and centrality analysis

As can be seen in Fig 3, it can be seen that the factor with the greatest degree of influence is the level of regional economic development (A6), indicating that A6 has the greatest comprehensive influence on other factors, while the factor with the greatest degree of being influenced is the net per capita income of farmers (A11), indicating that A11 is the factor with the greatest degree of being influenced. As can be seen in Fig 4, the factor with the greatest centrality is still the level of regional economic development (A6), which indicates that this factor plays the greatest role in the system and is the most important factor in the system. As can be seen in Fig 2, the weight value of A6 reaches 0.09, ranking first.

The top five factors influencing the sustainable development of plateau characteristic are, in descending order, agricultural capital input (A1), financial support (A15), agricultural machinery mechanization level A9, labor quality (A3), and pesticide use per hectare (A14). Factors

with high positive causes can easily influence other factors, therefore, when controlling the factors of sustainable development of plateau characteristic, attention should be paid to the above-mentioned factors with high positive causes. Therefore, when controlling the factors of sustainable development of plateau characteristic agriculture, we should focus on the management of the above-mentioned factors with high positive causes.

The top five influencing factors of sustainable development of agriculture with plateau characteristic from big to small (absolute value) are farmers' per capita net income (A11), agricultural product price index (A2), insurance support (A17), brand building (A16) and qualified rate of agricultural products (A7). The factor of high negative cause degree is easily affected by other factors. Therefore, when controlling the factor of high negative cause degree, we should also pay attention to the control of the upstream factors connected with it.

The order of the weight value of the influencing factors of sustainable development of agriculture with plateau characteristic from big to small is regional economic development level (A6), per capita net income of farmers (A11), brand building (A17), scientific and technological support (A18) and the number of agricultural enterprises and organizations (A8). It shows that the above elements play an important role in the system and are important elements in the system. Combined with the further analysis of the centrality degree and cause degree map, it can be seen that among the five elements, the regional economic development level (A6), the scientific and technological support (A18), and the number of agricultural enterprises and organizations (A8) are relatively high, because they are considered as the cause factors. When formulating relevant measures, priority should be given to these elements. On the other hand, the causes of farmers' per capita net income (A11) and brand building (A17) are relatively low, so they are considered more as a result. The further analysis combined with the AISM adversarial diagram shows that the A11 and A17 elements are located in the L1 and L2 layers, which are important result elements. A6, A8, and A18 belong to the middle layer and play the role of a connecting link between the preceding and the next. When formulating relevant measures, we should focus on considering the influence of other elements on these three elements so as to promote them to exert the greatest positive influence.

## 5. Analysis of sustainable development status of plateau agriculture in Yunnan Province

According to the analysis of the fourth part of this paper, the sustainable development of plateau agriculture in Yunnan Province should focus on agricultural capital investment A1, labor force quality A3, transportation cargo turnover A5, level of mechanization of agricultural machinery A9, pesticide use per hectare A14, insurance support A16, brand building A17, and science and technology support A18. To this end, we should formulate relevant support and security policies, and increase the corresponding capital investment. Since 2012, Yunnan Province has taken a variety of measures to develop modern agriculture with plateau characteristic on the basis of market demand and relying on the advantages of resources. In the aspects of transportation construction, green production, scientific research investment, and insurance guarantee, a series of specific measures have been formulated and implemented in practical work, which has achieved remarkable results and promoted the sustainable development of plateau characteristic agriculture in Yunnan Province. The specific work is as follows: (1) Yunnan Province has actively accelerated infrastructure construction and formulated a series of measures, including promoting the construction of infrastructure such as road networks, aviation networks, and water networks, and speeding up the construction of major international channels. These efforts have enabled Yunnan's agricultural products to be sold to more than 150 cities and more than 40 countries and regions across the country. In order to improve the

transport capacity of agricultural products, the comprehensive transportation investment in Yunnan Province reached 219.613 billion yuan in 2018, ranking first in the country. In addition, the number of navigation points in Southeast Asia and South Asia in Yunnan Province also ranks first in the country [33], providing a guarantee for the transportation of agricultural materials and agricultural products.

(2) To improve the quality, efficiency, and competitiveness of modern agriculture with highland characteristics, strengthen the governance of the outstanding agricultural environment, and provide a good environmental foundation for the green transformation of agricultural development. To this end, Yunnan Province has intensively implemented actions to reduce fertilizers and pesticides and increase their efficiency, maintaining negative growth in the use of fertilizers and pesticides. Between 2009 and 2016, the amount of fertilizers applied to Yunnan agriculture increased from 1,713,900 tons to 2,355,800 tons, and has since decreased year by year to 1,966,500 tons. Pesticide application per hectare, on the other hand, increased from 42,600 tons in 2009 to 58,600 tons in 2015, and subsequently decreased year by year. By 2020, the amount of pesticides applied per hectare has been reduced to 44,800 tons. On the premise of ensuring high agricultural production and good harvest, the use of chemical fertilizers and pesticides has been reduced, which fundamentally ensures the production environment and product quality of highland agricultural specialties.

(3) The formulation of policies and measures to actively support the entrepreneurial development of agricultural enterprises, the development of high-quality agricultural leading enterprises, and the improvement of the level of agricultural machinery mechanization have also led to an increase in farmers' employment and income, allowing farmers to share more of the value-added benefits of agricultural industries with plateau characteristic. From 2009 to 2020, the number of agricultural enterprises and organizations in Yunnan Province increased from 7026 to 97737, and the level of agricultural machinery mechanization increased from 1.29 to 2.27. Over the past decade, the per capita disposable income of rural residents in Yunnan Province has grown at an average annual rate of 9.1%, of which operating income accounts for 48.4%, which is about 14 percentage points higher than that of the whole country, and the proportion of the illiterate labor force has dropped from 8.34% to 5.78%. Among them, agriculture with plateau characteristic has played a great role.

(4) To deepen the reform in the field of agricultural science and technology with the structural reform on the supply side of agriculture as the main line, explore and develop new models of agricultural science and technology such as the science and technology commissioner system, and construct the support and guarantee of agricultural insurance. 2009 to present, the insured amount of agricultural insurance in Yunnan Province has increased from 5.701 billion yuan to 153.341 billion yuan, which is 25.9 times. Financial investment in agricultural science and technology research increased from 1.306 billion yuan to 6.494 billion yuan, and the number of agricultural science and technology activities accounted for no less than 80% of the number of agricultural research and comprehensive services for a long time.

## 6. Conclusions and recommendations

This paper takes the influencing factors of the sustainable development of plateau characteristic agriculture in Yunnan Province as the research object, firstly uses DEMATEL-AISM to establish an analysis model, clarifies the main influencing factors of the sustainable development of plateau characteristic agriculture in Yunnan Province, and analyzes and discusses the relationship between the main influencing factors. Established a set of plateau characteristic agricultural sustainable development influencing factors identification and evaluation system. Then, an empirical study was conducted to reveal the main factors and changes affecting the

sustainable development of plateau characteristic agriculture in Yunnan Province from 2009 to 2020, using the sustainable development of plateau characteristic agriculture in Yunnan Province as an example. The analysis shows that the judgment matrix established by this method can better synthesize subjective and objective evaluation opinions, and can reflect the structural relationship and strength of the relationship between the main factors influencing each other, which meets the needs of evaluation and judgment. It provides a theoretical basis and reference for the policy formulation and development planning of sustainable development of plateau characteristic agriculture in Yunnan Province.

Based on the above study, the following recommendations are made to promote the sustainable development of plateau characteristic agriculture in Yunnan Province:

1. Establish a correct development concept of sustainable development of plateau characteristic agriculture. Should fully reflect the requirements of green, ecological priority, in-depth implementation of chemical fertilizer, pesticide reduction, and efficiency action, to maintain the negative growth in the use of chemical fertilizers, and pesticides. To promote the sustainable development of plateau characteristic agriculture and resource conservation, environmental protection, and organic combination of plateau agriculture, to promote the "green transformation". At the same time to create advantageous characteristics of the brand of agricultural products, the establishment of high-quality, efficient, and safe branding system of agricultural products, improve brand value, and expand the influence of agricultural products with plateau characteristic.

2. Further strengthen the construction of water conservancy, transportation, and other infrastructures. Perfect infrastructure construction is the foundation to guarantee the smooth development of plateau characteristic agriculture. In recent years, Yunnan Province has made great progress in the construction of highways, railroads, and airlines, but the infrastructure construction in Yunnan Province still needs to be further strengthened. The government should increase financial support to improve the construction of water conservancy, logistics, and other facilities to guarantee the steady and rapid development of plateau characteristic agriculture.

3. Continue to increase the financial support for scientific research and human capital. The government should invest more money, increase the investment in agricultural research, improve the level of scientific and technological content of plateau characteristic agricultural production, and enhance the market competitiveness of characteristic agricultural products; pay attention to the investment in the human capital of plateau characteristic agriculture, pay attention to education, strictly implement the nine-year compulsory education, improve the level of rural knowledge, and also increase the knowledge of agricultural Publicity, training, etc. to improve human capital.

4. Improve and perfect the existing agricultural insurance model. The prospect of developing local specialty agriculture in Yunnan Province is broad, and a flexible, sound reasonable, and perfect agricultural insurance model should be formed according to the development needs of plateau characteristic industries, so as to break the blockage in production, processing and circulation and smooth the whole process of agricultural production. The government should vigorously guide the high-quality development of the insurance industry and improve the service capacity of agricultural insurance. Insurance institutions should develop reasonable agricultural insurance products with plateau characteristic to meet the multi-level, differentiated, and broad coverage risk protection needs of farmers, and realize a new situation of win-win development for the government, insurance institutions, enterprises, and farmers.

To sum up, this paper uses many factors that affect the sustainable development of agriculture with plateau characteristic, and first uses the DEMATEL model to calculate the influence degree, center degree, cause degree, and other index values of various factors. Then, based on the degree centrality of DEMATEL and the transitivity principle of influence, the decision matrix is obtained by the operation of the center degree and cause degree, and the adjacency matrix is obtained by partial order calculation of the decision matrix. Thirdly, the reachability matrix is obtained, the antagonistic hierarchy is divided according to the result priority and cause priority rules, and the AISM model of causal reachability level is derived. The model shows that traffic, environment, and insurance support factors are the root factors affecting the sustainable development of agriculture with plateau characteristic, while the regional economy, scientific and technological support, and the development of agricultural enterprises are of high importance. Financial support, the level of agricultural mechanization, the quality of labor, and other factors can not be ignored, and when these basic factors are effectively improved, they will lead and promote the effect at all levels. To achieve the ultimate goal. This conclusion has also been verified in the analysis of the state of agricultural development in the Yunnan plateau over the years. Compared with the traditional quantitative and qualitative analysis models, the combined use of the two DEMATEL-AISM models can comprehensively analyze the problem from different angles and has the advantages of a more comprehensive model, more accurate analysis results, more efficient decision-making process and better response to complex problems, so it is a better analysis model. However, in the future, we need to further verify the effectiveness of the DEMATEL-AISM association method and strengthen the application of this method in green innovation.

## Supporting information

**S1 File. Raw statistics for analysis.** Statistics on factors influencing the sustainable development of plateau characteristic agriculture from 2009 to 2020.
(XLSX)

**S2 File. Model derivation data and results.** The intermediate calculation process data and the final results obtained by the model calculation.
(XLSX)

## Author Contributions

**Conceptualization:** Hai Liu.

**Data curation:** Wei Wang, Pengfei Zhao, Mo Han.

**Formal analysis:** Hai Liu.

**Investigation:** Wei Wang.

**Methodology:** Hai Liu.

**Resources:** Wei Wang, Mo Han.

**Software:** Hai Liu.

**Validation:** Wei Wang.

**Visualization:** Pengfei Zhao.

**Writing – original draft:** Hai Liu.

**Writing – review & editing:** Hai Liu.

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
