## [Decision Letter · Decision Letter 0]

6 Mar 2023

PONE-D-23-00627Research on the Influence Factors of Sustainable Development of Plateau Characteristic Agriculture Based on DEMATEL and AISM Combined ModelPLOS ONE

Dear Dr. Liu,

Thank you for submitting your manuscript to PLOS ONE. After careful consideration, we feel that it has merit but does not fully meet PLOS ONE’s publication criteria as it currently stands. Therefore, we invite you to submit a revised version of the manuscript that addresses the points raised during the review process.

We look forward to receiving your revised manuscript.

Kind regards,

Mihajlo Jakovljevic, MD, PhD, MAE

Academic Editor

PLOS ONE

Journal Requirements:

2. Our internal editors have looked over your manuscript and determined that it is within the scope of our Sustainability and the Circular Economy Call for Papers. The Collection will encompass a diverse and interdisciplinary set of submissions related to sustainability and the circular economy, focusing on production models, business plans, and the contribution of global initiatives to increased sustainability in economic, environmental, and social terms. Additional information can be found on our announcement page: Sustainability and the Circular Economy - PLOS Collections . If you would like your manuscript to be considered for this collection, please let us know in your cover letter and we will ensure that your paper is treated as if you were responding to this call. If you would prefer to remove your manuscript from collection consideration, please specify this in the cover letter.

6. Please ensure that you refer to Figure 2 in your text as, if accepted, production will need this reference to link the reader to the figure.

7. We note you have included a table to which you do not refer in the text of your manuscript. Please ensure that you refer to Table 3,4 and 5 in your text; if accepted, production will need this reference to link the reader to the Table.

Reviewers' comments:

Reviewer's Responses to Questions

**Comments to the Author**

1. Is the manuscript technically sound, and do the data support the conclusions?

Reviewer #1: Partly

Reviewer #2: Yes

Reviewer #3: Partly

2. Has the statistical analysis been performed appropriately and rigorously? 

Reviewer #1: Yes

Reviewer #2: Yes

Reviewer #3: No

3. Have the authors made all data underlying the findings in their manuscript fully available?

Reviewer #1: Yes

Reviewer #2: No

Reviewer #3: No

4. Is the manuscript presented in an intelligible fashion and written in standard English?

Reviewer #1: No

Reviewer #2: Yes

Reviewer #3: Yes

5. Review Comments to the Author

Reviewer #1: The reviewer believes that the topic “Research on the Influence Factors of Sustainable Development of Plateau Characteristic Agriculture Based on DEMATEL and AISM Combined Model” is worthy of investigation. This paper has the potential to be accepted, but some important points have to be clarified or fixed before we can proceed and positive action can be taken.

Your abstract should clearly state the essence of the problem you are addressing, what you did and what you found and recommend.

Please specify the source of the data.

The language of this manuscript is very bad and needs help from native speakers.

Please review appropriate literature in the Introduction, with the research question clearly arising from that review.

The introduction P3, LINE67-77. This section should explain the study's context and research objective. Furthermore, the research gap needs to be narrowed after analyzing the previous studies. The research method is not adequately explained in the first section.

P3, LINE78-86. This a very vague statement. These sentences do not provide any information on how the concept could be conceptualized? - The Introduction should have 1) a concise but complete justification of the topic's importance both academically and practically, and 2) an explanation of the gaps both in research and practice. Please review appropriate literature in the Introduction, with the research question clearly arising from that review.

-The manuscript does not answer the following concerns: Why is it timeliness to explore such a study? What makes this study different from the previously published studies? Are there any similarly findings in line with the previously published studies? Are the findings different from prior academic studies that were conducted elsewhere, if any? See the following: Developing a conceptual partner matching framework for digital green innovation of agricultural high-end equipment manufacturing system toward agriculture 5.0: A Novel Niche Field Model Combined With Fuzzy VIKOR. Frontiers in Psychology, 2022; 13: 924109. https://doi.org/10.3389/fpsyg.2022.924109

There is no flow in the text. It partly depends on the lack of proofreading but also on the fact that many statements and claims are made without being followed up by a clear and logical discussion.

-More importantly, the choice of the variables should be explained in light of the theory and the prior literature on the topic.

See the following: Evolution of Agricultural Innovation Ecosystem in County Areas: A Life-Cycle Perspective of Cases in Hebei Province", Mathematical Problems in Engineering, vol. 2022, Article ID 5262248, 21 pages, 2022. https://doi.org/10.1155/2022/5262248

I suggest authors here clearly explain the model building process, and what previous studies have used similar models.

Methodology: Model.. I suggest authors here build your main heading on Research and data methodology. Clearly explain the model building process, and what previous studies have used similar models (model testing approach).

The authors should emphasize the important role of digital technology in green innovation in future research. Please consider this structure for manuscript final part.

-Discussion

-Conclusion

-Managerial Implication

-Practical/Social Implications

-Discussion needs to be a coherent and cohesive set of arguments that take us beyond this study in particular, and help us see the relevance of what authors have proposed. Authors should create an independent “Discussion” section. Author need to contextualize the findings in the literature, and need to be explicit about the added value of your study towards that literature. Also other studies should be cited to increase the theoretical background of each of the method used. Findings should be contextualized in the literature and should be explicit about the added value of the study towards the literature. Limitations and future research.

Please make sure your conclusions' section underscores the scientific value-added of your paper, and/or the applicability of your findings/results. Highlight the novelty of your study. In addition to summarizing the actions taken and results, please strengthen the explanation of their significance. It is recommended to use quantitative reasoning comparing with appropriate benchmarks, especially those stemming from previous work. See the following: An adoption-implementation framework of digital green knowledge to improve the performance of digital green innovation practices, https://doi.org/10.1016/j.jclepro.2022.132608

Reviewer #2: Recommendations:

- Can you clearly include the starting hypotheses and if these are fulfilled later?

- What lines of future research can be drawn?

- Why is it important to know the research on this topic?

- The discussion and the conclusion should be improved.

- Present clearly what possible implications or applications your findings have.

Reviewer #3: This paper established an analytical model using DEMATEL-AISM to clarify the main influencing factors on the sustainable development of plateau characteristic agriculture in Yunnan Province and conducted an empirical study. I think this paper is interesting. I thank the authors. However, the manuscript does not meet the publishing requirements at present, and the following key problems need to be solved.

Comment 1: The title of the figure should be centered.

Comment 2: Figure 1 shows the number 0 instead of the letter O mentioned in the text. It is also suggested to replace the expression “D” of the decision matrix to distinguish it from the degree of influence “D” or indicate what the current “D” is when expressing it later.

Comment 3: It is suggested to divide the steps of DEMATEL and AISM into two small frameworks in Figure 1 and divide them accordingly in the subsequent analysis, so that readers can understand the basic steps of the above two methods through reading this article.

Comment 4: Table 3 has problems with the direct impact matrix. The direct impact matrix should be a square matrix, that is, according to the number of factors studied in this text, it should be 18 × 18.

Comment 5: In order to increase the readability of the article, it is suggested to add the matrix to be calculated in the titles of 3.2.1 and 3.2.2, just like the title of 3.2.3.

Comment 6: It is suggested to explain the first parameter in the formula under the corresponding formula.

Comment 7: Figures 2, 3 and 4 are followed by no corresponding text description. If the author wants to make an analysis in Section 4, it is suggested to specify the figure in the order of Section 3 for explanation, so as to enhance the organization and logic of the article.

Comment 8: Decision matrix D, as a bridge between DEMATEL and AISM, is not reflected in Section 3.

Comment 9: There are some basic errors. For example, the title of 4.1 and 4.2 is incorrect (the word "analysis" is separated by spaces). The corresponding analysis of lines 204 and 221 should be the interpretation of Figure 5 (Figure 4 is written in the text). Line 223 refers to L7 level (the corresponding highest level in the text is L6).

Comment 10: Lines 247 and 256 are explained in Figure 4, but there is no difference of the explanations between centrality and causation. The author is suggested to further polish the full text.

Comment 11: Section 5 is not relevant to the previous analysis and is too long. It is recommended to merge Section 5 and Section 6 and reduce the length of Section 5.

6. PLOS authors have the option to publish the peer review history of their article (what does this mean?). If published, this will include your full peer review and any attached files.

Reviewer #1: No

Reviewer #2: No

Reviewer #3: No

---

## [Author Response · Author response to Decision Letter 0]

11 Jul 2023

We have made corresponding revisions and replies to the revised opinions of the three reviewers as required, for details, please refer to the specific content in the "ResponsetoReviewers.docx" file

---

## [Decision Letter · Decision Letter 1]

4 Oct 2023

PONE-D-23-00627R1Research on the Influence Factors of Sustainable Development of Plateau Characteristic Agriculture Based on DEMATEL and AISM Combined ModelPLOS ONE

Dear Dr. Liu,

Thank you for submitting your manuscript to PLOS ONE. After careful consideration, we feel that it has merit but does not fully meet PLOS ONE’s publication criteria as it currently stands. Therefore, we invite you to submit a revised version of the manuscript that addresses the points raised during the review process.

 Please submit your revised manuscript by Nov 18 2023 11:59PM. If you will need more time than this to complete your revisions, please reply to this message or contact the journal office at plosone@plos.org. Please include the following items when submitting your revised manuscript:A rebuttal letter that responds to each point raised by the academic editor and reviewer(s). You should upload this letter as a separate file labeled 'Response to Reviewers'.A marked-up copy of your manuscript that highlights changes made to the original version. You should upload this as a separate file labeled 'Revised Manuscript with Track Changes'.An unmarked version of your revised paper without tracked changes. You should upload this as a separate file labeled 'Manuscript'.If applicable, we recommend that you deposit your laboratory protocols in protocols.io to enhance the reproducibility of your results. Protocols.io assigns your protocol its own identifier (DOI) so that it can be cited independently in the future. For instructions see: https://journals.plos.org/plosone/s/submission-guidelines#loc-laboratory-protocols. Additionally, PLOS ONE offers an option for publishing peer-reviewed Lab Protocol articles, which describe protocols hosted on protocols.io. Read more information on sharing protocols at https://plos.org/protocols?utm_medium=editorial-email&utm_source=authorletters&utm_campaign=protocols.

We look forward to receiving your revised manuscript.

Kind regards,

Mihajlo Jakovljevic, MD, PhD, MAE

Academic Editor

PLOS ONE

Journal Requirements:

Reviewers' comments:

Reviewer's Responses to Questions

**Comments to the Author**

1. If the authors have adequately addressed your comments raised in a previous round of review and you feel that this manuscript is now acceptable for publication, you may indicate that here to bypass the “Comments to the Author” section, enter your conflict of interest statement in the “Confidential to Editor” section, and submit your "Accept" recommendation.

Reviewer #1: (No Response)

Reviewer #3: (No Response)

2. Is the manuscript technically sound, and do the data support the conclusions?

Reviewer #1: (No Response)

Reviewer #3: Yes

3. Has the statistical analysis been performed appropriately and rigorously? 

Reviewer #1: (No Response)

Reviewer #3: I Don't Know

4. Have the authors made all data underlying the findings in their manuscript fully available?

Reviewer #1: (No Response)

Reviewer #3: No

5. Is the manuscript presented in an intelligible fashion and written in standard English?

Reviewer #1: (No Response)

Reviewer #3: Yes

6. Review Comments to the Author

Reviewer #1: The introduction section was not sufficiently modified. I cannot identify the improvements according to my comments.

Reviewer #3: I would like to thank the authors for their efforts. However, some errors in the manuscript still need to be corrected by the author.

Comment 1: Please confirm whether the full name of AISM is "antagonistic interpretation structure model" or "Adversarial Interpretive Structure Modeling Method".

Comment 2: The full name of DEMATEL and AISM were not given when they first appeared in the text.

Comment 3: Adjust the font size of Table 3 appropriately to align the table content.

Comment 4: There is no corresponding change from "the degree of influence" D to E below title 3.2.4, and there is also a similar error below Fig 3.

Comment 5: The adjacency matrix A calculation expression under heading 3.2.7 should be placed under heading 3.2.8.

Please carefully review and correct any fundamental errors in the manuscript.

7. PLOS authors have the option to publish the peer review history of their article (what does this mean?). If published, this will include your full peer review and any attached files.

Reviewer #1: No

Reviewer #3: No

---

## [Author Response · Author response to Decision Letter 1]

15 Nov 2023

We have responded to the opinions of all the reviewers as required. Please refer to the underlined section for details. The following is the expert opinion and reply to the manuscript review:

Reviewers' Comment: Reviewers' Have the authors made all data underlying the findings in their manuscript fully available?.

Reviewer's Responses to Questions：The data such as mean, median and variance have been supplemented in the uploaded Statistics Data.xlsx file.

Review Comments to the Author:

Reviewer #3: I would like to thank the authors for their efforts. However, some errors in the manuscript still need to be corrected by the author.

Reviewers' Comment 1: Please confirm whether the full name of AISM is "antagonistic interpretation structure model" or "Adversarial Interpretive Structure Modeling Method".

Reviewer's Responses to Questions：Revised, AISM's full name is: "Adversarial Interpretive Structure Modeling Method"

Reviewers' Comment 2: The full name of DEMATEL and AISM were not given when they first appeared in the text.

Reviewer's Responses to Questions：This paper has been revised and the full name is given when it appears for the first time.

Reviewers' Comment 3: Adjust the font size of Table 3 appropriately to align the table content.

Reviewer's Responses to Questions：The text has been modified to align the contents of the table.

Reviewers' Comment 4: There is no corresponding change from "the degree of influence" D to E below title 3.2.4, and there is also a similar error below Fig 3.

Reviewer's Responses to Questions：The identification name of influence degree E has been revised to unify it.

Reviewers' Comment 5: The adjacency matrix A calculation expression under heading 3.2.7 should be placed under heading 3.2.8.

Reviewer's Responses to Questions：Formula 7 in 3.2.7 is the calculation method of decision matrix D, which is explained in 3.2.7.

---

## [Editor Report · Decision Letter 2]

11 Jan 2024

Research on the Influence Factors of Sustainable Development of Plateau Characteristic Agriculture Based on DEMATEL and AISM Combined Model

PONE-D-23-00627R2

Dear Dr. Liu,

We’re pleased to inform you that your manuscript has been judged scientifically suitable for publication and will be formally accepted for publication once it meets all outstanding technical requirements.

Kind regards,

Mihajlo Jakovljevic, MD, PhD, MAE

Academic Editor

PLOS ONE

Additional Editor Comments (optional):

Dear Authors,

Dear Plos One Editorial Office,

Based on the two rounds of meaningful but mutually not so consistent external peer reviews and my own Editorial insight into manuscript revision adopting most of provided recommendations, hereby I render a Decision:

Accept without further changes

Congratulations !

Mihajlo (Michael) Jakovljevic M.D. Ph.D. MAE

UNESCO - The World Academy of Sciences (TWAS) Academician

Editor-in-Chief, Cost-effectiveness and resource allocation (CERA), BMC

Full Professor, Founding Head of Department Global Health Economics & Policy

University of Kragujevac Faculty of Medical Sciences, SERBIA

Full Professor, Institute of Comparative Economic Studies

Hosei University Faculty of Economics, Tokyo, JAPAN

On Google Scholar: http://scholar.google.com/citations?user=4KELK8wAAAAJ&hl=en&oi=ao

https://resource-allocation.biomedcentral.com/
---

## [Editor Report · Acceptance letter]

30 Jan 2024

PONE-D-23-00627R2 

PLOS ONE

Dear Dr. Liu, 

I'm pleased to inform you that your manuscript has been deemed suitable for publication in PLOS ONE. Congratulations! Your manuscript is now being handed over to our production team.

Kind regards, 

on behalf of

Professor Mihajlo Jakovljevic 

Academic Editor

PLOS ONE